# Remote Sensing Approaches for Meteorological Disaster Monitoring: Recent Achievements and New Challenges

**DOI:** 10.3390/ijerph19063701

**Published:** 2022-03-20

**Authors:** Peng Ye

**Affiliations:** 1Urban Planning and Development Institute, Yangzhou University, Yangzhou 225127, China; 007839@yzu.edu.cn; Tel.: +86-156-5195-8693; 2College of Architectural Science and Engineering, Yangzhou University, Yangzhou 225127, China

**Keywords:** meteorological disasters, remote sensing monitoring, monitoring task arrangement and organization, meteorological disaster information extraction, multi-temporal disaster information change detection

## Abstract

Meteorological disaster monitoring is an important research direction in remote sensing technology in the field of meteorology, which can serve many meteorological disaster management tasks. The key issues in the remote sensing monitoring of meteorological disasters are monitoring task arrangement and organization, meteorological disaster information extraction, and multi-temporal disaster information change detection. To accurately represent the monitoring tasks, it is necessary to determine the timescale, perform sensor planning, and construct a representation model to monitor information. On this basis, the meteorological disaster information is extracted by remote sensing data-processing approaches. Furthermore, the multi-temporal meteorological disaster information is compared to detect the evolution of meteorological disasters. Due to the highly dynamic nature of meteorological disasters, the process characteristics of meteorological disasters monitoring have attracted more attention. Although many remote sensing approaches were successfully used for meteorological disaster monitoring, there are still gaps in process monitoring. In future, research on sensor planning, information representation models, multi-source data fusion, etc., will provide an important basis and direction to promote meteorological disaster process monitoring. The process monitoring strategy will further promote the discovery of correlations and impact mechanisms in the evolution of meteorological disasters.

## 1. Introduction

As an important source of potential damage, natural disasters have long received universal attention in the field of disaster prevention and reduction. The mitigation of natural disasters is a key factor in ensuring people’s livelihood. According to a sigma report published by the Swiss Re Institute, the total global economic losses from natural disasters reached 105 billion USD in 2021, again exceeding the average for the previous 10 years [1]. Meteorological disasters are one of the most important types of natural disaster. According to the statistics of the World Meteorological Organization, more than 90% of natural disasters are directly or indirectly caused by meteorological conditions [2]. As meteorological disasters include various types of events, extensive regional occurrence, and obvious seasonal characteristics, they have serious impacts on the ecological environment, economic development, social stability and national security, and higher requirement for the rapid and effective prevention and control of emergency management sectors [3,4]. Therefore, the research on meteorological disasters and related issues is still an important research frontier in the field of international disaster prevention and reduction [5].

In recent years, along with the rapid development of sensor technology, aerospace platform technology, and data communication technology, the global observation capability of satellite, aircraft, and land integration has been greatly enhanced [6,7,8]. Remote sensing data usually contain two major characteristics of spectral and spatial structures, which can effectively describe the shape distribution and morphological structure information of spatial objects on the Earth’s surface [9]. By detecting changes in the spectral and spatial structure of the corresponding regions in the multi-temporal remote sensing data, it is possible to identify the type of the disaster, determine its intensity, and analyze its scope of influence [10]. Compared with traditional disaster monitoring methods, remote sensing monitoring has obvious advantages in terms of timeliness, space, and economy and has been widely used in many fields of disaster management [11]. Therefore, remote sensing technology not only improves the objectivity and accuracy of disaster monitoring but also enables a comprehensive understanding of the disaster process.

According to the fifth Climate Report of the United Nations Intergovernmental Panel on Climate Change (IPCC), climate warming will intensify in the future [12]. At the 26th Conference of the Parties to the United Nations Framework Convention on Climate Change, the World Meteorological Organization also issued the “State of the Global Climate 2021”, which identified extreme weather events as a major threat to global warming [13]. Employing remote sensing, the continuous spatio-temporal information of terrain distribution, factor status, and historical evolution can be obtained, which helps realize large-scale, high-time-efficiency and high-precision meteorological disaster monitoring [14]. This paper is focused on answering the following research questions:(1)What are the objectives of meteorological disaster process monitoring?(2)How are remote sensing approaches being used for meteorological disaster process monitoring?(3)What are the current gaps in remote sensing-based meteorological disaster process monitoring?

The rest of the paper is organized as follows: Section 2 describes the basic ideas of meteorological disaster process monitoring; Section 3 presents the remote sensing monitoring approaches; Section 4 discusses the open problems and challenges in the future. The paper closes with a conclusion in Section 5.

## 2. Basic Ideas of Meteorological Disaster Process Monitoring

Disaster is a general term for objects that have destructive effects on human beings and the environment on which human beings depend for their survival [15]. As one type of disaster, meteorological disasters refer to the disasters caused by weather changes [16]. Different meteorological disasters are closely related to different weather phenomena, and their impacts and hazards are also different [17]. However, the occurrence of meteorological disasters is not only related to weather phenomena but also closely related to human beings and the surrounding environment [18,19]. For instance, a rainstorm of the same scale in a densely populated city may result in disasters such as urban waterlogging; if it occurs in an uninhabited sea, it is merely a weather phenomenon.

For the remote sensing monitoring of meteorological disasters, the aim is to meet the disaster management needs of disaster prevention and mitigation, and to obtain various pieces of disaster information in different periods in time [20]. Following the technical methods of remote sensing monitoring, firstly, it is necessary to arrange the monitoring tasks, including determining timescales, selecting data sources and monitoring information modeling. As far as meteorological disaster monitoring is concerned, the timescale refers to the monitoring timespan and the minimum time interval. Due to the different types of disasters or monitoring objects, the appropriate timescale will be different. Data sources’ selection refers to the determination of the types of sensors used for meteorological disaster monitoring. The selection basis will be influenced by the characteristics of different types of disasters and the size of timescales. Through monitoring information modeling, it is possible to determine the type of meteorological disaster information that needs to be monitored, and also to clarify the organization of various types of monitoring information. This provides support for the mission execution of the remote sensing monitoring of meteorological disasters and the later information analysis.

After determining the arrangement for the remote sensing monitoring of meteorological disasters, it is possible to follow the relevant requirements to perform monitoring tasks and begin to extract various types of disaster information from remote sensing data. Disaster is the product of the comprehensive effect of three factors: pregnant environment, causing factor, and disaster-bearing body [21,22]. For meteorological disasters, the pregnant environment refers to the comprehensive earth surface environment, composed of the atmosphere, lithosphere, and hydrosphere; causing factor refers to weather phenomena that may cause property loss, casualty, resource and environment damage, and social disorder; disaster-bearing body refers to the main bodies of human society that are affected and damaged by disasters, including human beings themselves and all aspects of social development [23,24].

Although different meteorological disasters present different patterns and ever-changing situations in disaster factors, the occurrence of meteorological disasters is always a gradual, evolutionary process [25,26,27]. In the process of one meteorological disaster, the degree to which the natural environment was affected was first aggravated and then gradually restored to a normal state [28]. Moreover, when one meteorological disaster is over, the next one will start, and the natural cycle will repeat itself. Therefore, meteorological disaster has continuity, and all meteorological disaster factors (pregnant environment, causing factor, and disaster-bearing body) are in the process of mutual influence and constant change. The monitoring of meteorological disasters is not only limited to the extraction of disaster information at a certain time, but also to observing the change in disaster information on a time series.

The roadmap of meteorological disaster process monitoring based on remote sensing is shown in Figure 1. The monitoring of meteorological disasters focuses on the weather phenomena that cause disasters and the destructive consequences of disasters. When arranging the monitoring task, considering the monitoring requirements of different types of meteorological disasters, remote sensing data types, including optical image, radar, LiDAR, etc., are selected according to specific timescale and sensor characteristics. In addition, to clarify the types and organization of the information extracted from remote sensing, the information representation model of meteorological disasters is constructed. During disaster information extraction, remote sensing technologies, such as parameter inversion, target detection, and scene classification, are used to extract the information of the pregnant environment, causing factor, and the disaster-bearing body. During disaster information change detection, multi-temporal remote sensing data are registered, and the difference in disaster information is detected based on the change-detection method. The spatio-temporal differences and dynamic changes in disaster areas, disaster degree, and rescue activities during meteorological disasters can be further analyzed.

## 3. Remote Sensing Monitoring Approaches

### 3.1. Monitoring Task Arrangement and Organization

In the monitoring of meteorological disasters, it is first necessary to reasonably and comprehensively determine the monitoring tasks to clarify the monitoring timescale, carry out sensor planning and construct the conceptual model of monitoring information. The accurate representation of monitoring tasks is the basis and premise of accurately and effectively obtaining disaster information.

#### 3.1.1. Timescale Determination in Monitoring

As an important branch of geography, the meteorological disaster system has the dual attributes of “natural” and “social”, and it is also “scale” dependent on nature. On the one hand, the dependence is manifested in the significant timescale effect of disaster factors. Table 1 summarizes the appropriate observation timescale for different meteorological disaster factors [29,30,31]. On the other hand, the dependence is reflected in disaster measurement and monitoring methods, and there are also significant timescale effects [32,33]. For instance, it is necessary to choose remote sensing data with a short cycle for monitoring typhoons with drastic change. Therefore, the timescale is one of the original characteristics of meteorological disasters.

With the introduction of the timescale, when conducting meteorological disaster monitoring, the time boundary to be monitored should first be determined according to the time extent, and the disaster monitoring is limited to a certain period [34]. To understand the detailed characteristics of disasters in a certain period, the temporal dynamic characteristics of meteorological disasters can be obtained according to the time granularity [35,36]. Only by selecting a reasonable and appropriate timescale can the effectiveness of and difference in meteorological disaster monitoring be truly identified [37]. In practical applications, the temporal extent and granularity should be considered when selecting a reasonable timescale for meteorological disaster monitoring. It must be noted that there is often more than one applicable scale in scale selection, and multi-scale analysis is often required in the research.

#### 3.1.2. Sensor Planning in Monitoring

There are three main types of sensor planning solutions: single sensor, multi-sensor in single platform, and multi-sensor in multiple platforms. For the planning of a single sensor, due to the small number of sensors, the data processing steps are relatively simple, and the data specifications between the multi-period data are unified [38]. For instance, Sivanpillai et al. studied rapid flood inundation mapping based on Landsat images [39]. However, due to the single sensor type, fewer types of disaster information can be monitored. When using a multi-sensor in a single platform, it refers to the remote sensing data collected by different sensors on the same platform. For instance, the operational land imager (OLI) data and the thermal infrared sensor (TIRS) data were contained in Landsat-8. Kumar et al. derived the flooded areas from Landsat 8 OLI-TIRS image [40]. Due to the limited number of sensors available on a single platform, it is also relatively easy to obtain information about the available sensor’s observation capabilities [41]. When using a multi-sensor on multiple platforms, because of the abundance of available data types, the types of disaster information that can be monitored are more diverse. For instance, Kyriou et al. also studied flood mapping, while the approach used data from active and passive remote sensing sensors such as Sentinel-1 and Landsat-8 [42]. In addition, the potential utility of multi-sensor data fusion for different phases of disaster management: vulnerability assessment, early warning systems, disaster mitigation, response, damage assessment and recovery are delineated [43]. The relevant methods are given in Section 3.2.2. However, the difficulty in using multi-source sensor data will increase to a certain extent, which is mainly reflected in the differences in data specifications (resolution, coordinate system, etc.), the inconsistency of the monitoring period, and the applicability of band fusion.

However, most sensor planning is based on the technical conditions of the sensor itself, which lacks the planning based on the meteorological disaster process monitoring; thus, this kind of planning method would not obtain detailed and specific solutions [44]. For example, it is difficult to directly obtain answers to questions such as “To monitor the status of a certain stage of a meteorological disaster, which combination of sensors is more suitable for current needs?”. Since each type of meteorological disaster has a unique evolutionary process, existing sensor planning methods cannot accommodate the long-term, continuous process monitoring requirements for a single disaster. Researchers are gradually focusing on sensor requirements for different meteorological disasters, analyzing the applicability of sensors according to three aspects: temporal, spatial, and spectral resolution. This paper is summarized in Table 2, according to the existing research results.

#### 3.1.3. Monitoring Information Modeling

In disaster management, the use of multi-period, multi-area, multi-source, and multi-department meteorological disaster monitoring information has gradually become normal. If there is no uniform standard for meteorological disaster information, it will be difficult to share information in disaster management, which is disadvantageous to efficient monitoring and emergency warning [51]. Therefore, the establishment of a reasonable meteorological disaster information representation model is the basic aim of meteorological disaster process monitoring.

The unified management of the distributed sensors and data resources by establishing metadata model specifications has become an important method for disaster information modeling. Considering the universality of the model, it is usually divided into the general metadata model and the dedicated metadata model. General metadata models include Common Alert Protocol (CAP), Emergency Data Exchange Language Distribution Element (EDXL-DE) and EDXL-rm [52]. The dedicated metadata model is specific to specific types of disasters, such as Tsunami Warning Markup Language (TWML) [53,54] and Cyclone Warning Markup Language (CWML) [55], associated with meteorological disasters. However, these specifications are biased towards the service of disaster warning, do not emphasize the importance of disaster monitoring information, and are not fully applicable to the process monitoring needs [56].

The knowledge representation method can provide semantic integration among people, heterogeneous systems, and people-to-systems. The following related techniques are gradually being introduced to disaster information modeling: 1) Ontology. Ontology is helpful in expressing the concept of disaster and eliminating semantic heterogeneity. After the foundation of standardized geographic ontology proposed by Kolas et al. [57], more specific geospatial ontologies have been established in disaster management, such as MDO, DOLCE+DnS, etc. [58,59]; 2) Knowledge graph. The knowledge graph can describe various entities and their complex relationships and is good at expressing various entities or events existing in the real world. The graph frameworks for natural disaster monitoring and assessment have been established and are currently used to represent information on typhoons, landslides, and their secondary disasters [60].

In summary, the existing disaster information models rarely consider the spatio-temporal factor; most of them record spatio-temporal information as common attributes [61]. A meteorological disaster system is a typical kind of structure and process, which is inherently bound to time and space. Disaster monitoring information is closely related to the disaster evolution process. It is necessary to consider the time and space characteristics in the disaster monitoring information model to support the further formal description and in-depth analysis of disaster monitoring information according to the spatio-temporal framework.

### 3.2. Meteorological Disaster Information Extraction

#### 3.2.1. Information Extraction of Pregnant Environment

The pregnant environment is the natural and human environment that causes disasters. The monitoring of the pregnant environment, mostly before the disaster occurs, and the monitoring results can help to evaluate the disaster risk and predict the disaster. For meteorological disasters, monitoring the pregnant environment is mainly carried out by the extraction of surface environmental information from the lithosphere to the atmosphere. There is a lot of information regarding the pregnant environment, such as altitude, slope, river, road, formation lithology, fault zone, etc., which can be divided into two categories: surface parameters and land cover.

##### Extraction of Surface Parameters

The methods of extracting surface parameters using remote sensing methods mainly include the empirical model method and the physical model method. The empirical model method is mainly implemented by establishing an empirical regression model between remote sensing observation signals and surface parameters [62]. To improve the precision of information extraction, the empirical model is constructed in the following two ways: (1) constructing the empirical model using the remote sensing data after the topographic correction; (2) direct use of topographic data (such as altitude, slope, aspect) to participate in the construction of empirical models [63]. However, the empirical model method is not strong in terms of spatio-temporal extension, and its applicability is often limited by time and space [64,65]. This is because the empirical model is based on a large number of remote sensing data characteristics and the corresponding statistical results of ground realities. The difference in weather and insolation at different times affects the characteristics of remote sensing data. The difference in land type in different spaces affects the statistical results of ground realities. Therefore, the empirical model built under a specific space-time is not suitable for a large-scale extension.

The main strategies for retrieving surface parameters based on physical models are as follows: (1) on the basis of considering the influence of local terrain on the imaging geometry, the surface parameters are extracted using a physical model that is suitable for a flat surface [66]; (2) on the basis of considering the spatial heterogeneity of the surface space, appropriately adjust the value range of the driving data of the existing flat physical model to make it more in line with the actual situation of the surface [67]; (3) develop physical models for specific terrain; this will also be the direction in which a great breakthrough may be made in the inversion modeling of special terrain surface parameters [68,69].

##### Extraction of Land Cover Information

To improve the accuracy of remote sensing extraction of land cover information, it is necessary to involve as much multi-source and multi-temporal remote sensing information and existing knowledge as possible, and also to develop scientific methods to effectively fuse heterogeneous information and eliminate redundancy [70,71]. A variety of methods have been developed to integrate multi-source information and can be summarized by the following three aspects: (1) based on the remote sensing image characteristics, such as spectral characteristics, shape characteristics, texture characteristics, and context characteristics, introduce the current mature artificial intelligence algorithms or mathematical methods, such as a decision tree model [72], Markov chain model [73], convolutional neural network model [74], etc., to develop a classification algorithm; (2) develop new classification algorithms to effectively coordinate multi-source and multi-temporal remote sensing data [75,76]; (3) introduce various thematic information and relevant geoscience knowledge into the research on land-cover classification with multi-source and heterogeneous information [77,78].

In recent years, with the acceleration of global urbanization, a great deal of infrastructure construction has rapidly changed land cover in a few decades or less [79]. The impact of urbanization on the changing pregnant environment, and even the occurrence of meteorological disasters, has gradually become one of the focuses of pregnant environment monitoring. Urbanization and urban consumption produce large amounts of greenhouse gases, which upset the balance of the earth’s ecosystem. This is a direct driver of climate change, contributing to global warming and to the deterioration of the pregnant environment in the ecosystem. Monitoring the pregnant environment under urbanization by remote sensing is an important reference to assess the risk of meteorological disasters [80]. For instance, Waghwala et al. assessed flood risk based on the satellite image of Resources-1, then they proposed that a change from a low to a high urbanization pattern is the main driver for increasing flood risk [81] (Figure 2).

#### 3.2.2. Information Extraction of Causing Factor

The cause of meteorological disasters is the weather phenomena that may cause disasters. The characteristic parameters of meteorological disasters can reflect the genetic characteristics, structural characteristics, and development process characteristics of disasters and are an important basis for the monitoring and evaluation of the causes. The monitoring of causes based on disaster-characteristic parameters focuses on the type of characteristic parameter on the one hand and on the method of extraction used for each characteristic parameter on the other hand. The extraction of information regarding the causes of different meteorological disasters is discussed in the following parts.

##### Drought

In principle, remote sensing approaches to drought monitoring can be divided into two categories: 1) the change in soil moisture can cause changes of soil spectral characteristics; 2) the change in plant physiological processes caused by drought can change the spectral properties of leaves and significantly affect the spectral properties of plant canopy. Sensors for drought remote sensing include a visible light band, near-infrared band, thermal infrared band, microwave band, etc. The common approaches include the soil thermal inertia method, soil moisture inversion method, vegetation index method, surface temperature method, and composite index method (Table 3).

**Table 3 ijerph-19-03701-t003:** Information extraction of drought.

Type	Index/Method	Description
Soil thermal inertia	Apparent thermal inertia (ATI) [82]	The higher the soil moisture content, the greater the soil thermal inertia, resulting in a small temperature difference between day and night.
Soil moisture	Microwave-based soil moisture retrieval [83]	The dielectric properties of liquid water are obviously different from those of dry soil.
Vegetation index	Vegetation condition index (VCI) [84]	Drought reduced the absorption of soil nutrients by vegetation and limited the growth of vegetation, resulting in changes in vegetation index.
	Anomaly vegetation index (AVI) [85]	
	Vegetation health index (VHI) [86]	
Surface temperature	Temperature condition index (TCI) [87]	Drought means that the soil water supply decreases, the surface temperature increases, the vegetation cover area will appear and the vegetation canopy temperature increases.
	Normalized difference temperature index (NDTI) [88]	
Composite index	Temperature vegetation drought index (TVDI) [89]	Considering the relationship between “vegetation index + land surface temperature” and drought. Due to the diversity of indexes, the inversion of these indexes usually depends on multi-source remote sensing data (Figure 3).
	Synthesized drought index (SDI) [90]	
	Optimized meteorological drought index (OMDI) and optimized vegetation drought index (OVDI) [91]	

##### Precipitation

Precipitation is the comprehensive result of atmospheric dynamic and thermal actions, which determines the external shape of precipitating cloud. Precipitation monitoring is mostly used to calculate the precipitation indirectly by detecting the cloud temperature or cloud particle information. This is not the same as directly observed precipitation but is currently one of the most difficult to observe atmospheric variables. Sensors for remote sensing of precipitation include visible light band, infrared band, active/passive microwave bands, and a combination of multiple sensors. Due to the different remote sensing data and retrieval algorithms, the precision of precipitation retrieval is different. The research focus has gradually turned to the fusion of precipitation products retrieved by multiple sensors to obtain more accurate precipitation information (Table 4).

##### Snow

Snow cover, as the most common upper boundary condition during freezing and thawing, directly affects the distribution and migration of soil moisture, soil temperature, freezing rate, freezing depth, etc. Snow cover has the unique spectral characteristics of a high reflection of visible light and a low reflection of short-wave infrared. Additionally, the snow particle size and the content of optical impurity affect the snow spectrum reflection. Sensors for the remote sensing of snow include a visible-light band, near-infrared band, far infrared band, and microwave band. Using the spectral characteristics, remote sensing approaches for snow cover monitoring can be divided into two categories: (1) binary snow identification (judging whether there is snow or not); (2) sub-pixel snow cover mapping in the form of area percentage, also known as snow coverage (Table 5).

##### Haze

Haze is a mixture of particles with different diameters. The scattering and absorption of electromagnetic waves change the radiative transmission process and affect the quality of remote sensing data. Remote sensing approaches to haze monitoring include four categories: spectral-feature-based method, image-preferential-based method, aerosol optical-thickness-based method, and multi-source remote sensing stereo-monitoring method (Table 6).

##### Typhoon

As the temporal and spatial resolution of the weather satellite has improved, satellite imagery has become the primary means of monitoring typhoons [115]. The satellite cloud image cannot provide quantitative information on wind field strength or a detailed image of the eye of the typhoon and its susceptibility to complex weather conditions such as clouds and fog [116]. Thus, correlation monitoring based on microwave data has strong application potential. For instance, Shen et al. studied the impact of assimilating radiances from the global satellite precipitation program (GPM) Microwave Imager (GMI) to track typhoon predictions. It was found that the data assimilation of clear-sky GMI radiance is able to depict the structure of typhoons by warming the typhoon inner core area [117]. The remote sensing approaches for typhoon monitoring are shown in Table 7.

##### Dust

Dust particles contain a lot of minerals and have strong backscattering to solar short-wave radiation. This changes the long-wave infrared radiation and seriously affects the visibility of the atmosphere. Thus, the spectral characteristics of dust are significantly different from those of clouds and the underlying surface [124]. Hyperspectral thermal infrared remote sensing has unique advantages in dust detection, which is significant for quantitative inversion of dust parameters. Especially when the concentration of dust in the atmosphere increases, the optical depth, height, and effective radius of dust particles can be quantitatively obtained in the infrared radiation signal. However, due to the slow change in dust aerosol scattering in the thermal infrared spectrum, there is a lot of redundancy in the hyperspectral thermal infrared data. The remote sensing approaches to dust monitoring are in Table 8.

#### 3.2.3. Information Extraction of Disaster-Bearing Body

For the disaster-bearing body, as the main body of human society that is directly affected and damaged by disasters, the core content of information extraction is used to analyze the damage to or recovery of personnel, social wealth, lifeline systems, etc. The target detection and scene classification approaches in remote sensing processing are mainly used to extract information on disaster-bearing bodies of different types.

##### Target Detection

For the extraction of information on the disaster-bearing body, the target object is all kinds of ground objects that are affected by meteorological disasters. According to the morphological characteristics of the object features, typical objects are generally classified into linear objects, blob objects, and complex objects. Table 9 combs the detection methods of common target types.

A large number of target detection methods have been accumulated, including statistical-based detection methods (such as threshold-based segmentation target detection, global threshold detection algorithm [131]; sliding-window-based adaptive threshold method; maximum likelihood detection) [132], based on structural analysis detection methods (such as the template matching method, mathematical morphology method) [133], model-based target detection (such as distribution model, visual saliency model, network model, etc.) and transform domain target detection (Hough transform, Radon transform, etc.) [134]. In recent years, visual saliency and manifold learning have had a better effect on target detection. The visual saliency model based on sparse coding and the multi-space target detection algorithm based on discriminant learning would cause fewer learning problems when training samples of hyperspectral target detection. The sparsely migrated, manifold, embedded feature extraction algorithm would effectively find the embedded low-dimensional feature sparse transform from the test data sample distribution information. In meteorological disasters, most disaster-bearing body types extracted by target detection are buildings, roads, and woodlands, and can be classified according to the severity of disaster (Table 10).

However, in the actual monitoring of meteorological disasters, the existing methods are usually highly targeted, lacking general and robust target detection models and algorithms. In a complex meteorological disaster environment, the false detection rate of the target detection method is too high for real-time applications. Additionally, regarding the target’s motion characteristics, visible cloud occlusion, and infrared background interference, it is necessary to carry out targeted improvement models and establish a highly applicable, multi-source, fusion, parallel, fast processing algorithm.

##### Scene Classification

The meteorological disaster recorded in the image itself is a complex scene, which is composed of multiple objects according to a certain spatial structure. Therefore, the scene classification method is suitable for the identification and classification of the image of the disaster-bearing body of meteorological disasters.

Scene classification is the domain of the whole-scene semantic understanding through extracting the visual features of the scene image, mapping the features, describing the content of the image, designing the classifier, and finally realizing the classification and recognition of the image scene. Scene classification mainly includes two key issues: image content description and classification decision. The image content description tries to obtain the most discriminative representation of the scene image, while the classification decision learns the training model of the training sample set, and models the calculation model that distinguishes certain scene categories from other scene categories. The algorithm used for scene classification has gone through three stages. The first is low-level features such as color, shape, and texture [147], which are classified by directly mapping these low-level features to high-level scene semantics. Secondly, the middle-level semantic features such as Gist, topic model, and Bag of Words (BoW) [148] are classified by shortening the semantic gap between low-level features and high-level scenes through middle-level semantic features. Thirdly, the machine learning models independently carry out data expression and feature extraction [149] and discard the pattern of the extracted features according to predetermined rules [150,151]; thus, they obtain improved classification results when applied to complex images. The commonly used machine learning methods include sparse coding [152], neural networks [153], SVMs [154,155] and deep learning [156]. The deep learning networks are composed of multiple nonlinear mapping layers, which represent a new method of intelligent pattern recognition, and are an important new direction in the field of remote sensing image processing [157]. In meteorological disasters, the most disaster-bearing body extracted by the scene method is the flood-submerging area affected by rainstorms or typhoons (Table 11).

Due to the complexity and diversity of the scene itself and the influence of various factors such as shooting angle and light, only relying on this single feature to classify the scene cannot meet the requirements of classification accuracy in practical applications. Some researchers began to use information fusion methods for scene classification. For instance, Wang et al. chose the corresponding sensors to collect the disaster scene information and proposed multi-sensor information fusion strategies for floods and some other disasters [165]. Regarding feature-level fusion, the fusion method based on serial connection is adopted. The feature-level fusion based on serial connection refers to the fusion of features into a new feature. Regarding decision-level fusion, the method based on a multi-classifier combination has been widely studied. However, most of these fusion methods only work at a certain level of information fusion, cannot make full use of the complementary information between different levels of fusion, and do not fully exploit the advantages of information fusion.

### 3.3. Multi-Temporal Disaster Information Change Detection

Meteorological disaster process monitoring is based on a series of spatio-temporal observation snapshots of a process-owned disaster, and is essentially a continuous observation of disaster information in multi-temporal remote sensing images [166]. Multi-temporal, remote sensing, image-change detection is the core method for the meteorological disaster process, including two stages of image registration and change detection.

#### 3.3.1. Remote Sensing Image Registration

The key to image registration is to find the optimal spatial mapping relationship between the reference image and the image to be matched, and to unify the image information that will be registered on the reference image to solve the inevitable geometric distortion in the image acquisition process [167]. In general, image registration is an important pre-processing task before change detection, especially for multi-sensor remote sensing data.

Geometric registration error is one of the most important sources of error in change detection. If high image registration accuracy is not obtained, a number of pseudo-variation regions will appear. However, for many feature-level change-detection methods, such as object-oriented methods, buffer analysis methods that consider registration errors can be used to compare extracted features or targets, thereby avoiding too harsh and highly accurate registration requirements [168]. Due to the influence of registration on change detection, image registration can be synchronized with change detection, making full use of unaltered object targets as the basis for image registration [169]. Image registration and change detection are solved as a whole, overcoming the transmission and accumulation of the registration error in the traditional method to improve the accuracy of change detection [170].

Radiation correction is another important aspect of preprocessing. Commonly used radiation correction methods include absolute radiation correction and relative radiation correction, and relative radiation correction is a more frequently used method because absolute radiation correction requires a large amount of imaging and other parameters [171]. The relative radiation correction method uses one image as a reference image to adjust the radiation characteristics of the other image to be consistent with the reference image, including image regression, pseudo-invariant feature, binary regression analysis, histogram matching, and so on [172]. The first three approaches are mostly based on the linear model, which assumes that there is some linear consistency in the overall difference in radiation between images. In fact, the difference in image radiation acquired by different types of sensors usually does not satisfy the linear model assumption. As the difference in radiation between images participating in the change detection is mainly reflected in the low-frequency part, the relative radiation correction method that considers the two-dimensional distribution of the low-frequency information of the image is often used in change detection.

#### 3.3.2. Remote Sensing Change Detection

Change detection aims to quantitatively analyze and determine the characteristics of surface change from multi-temporal remote sensing data. For meteorological disaster process monitoring, change detection can compare the state of the disaster, the affected area and the extent of the impact [173]. According to the level of the processing object, the change detection is divided into a pixel-level, an object-level, and a scene-level method.

Pixel-level change detection is suitable for the comparison and analysis of characteristic parameters of various meteorological disasters. The change-detection method based on the different images is the most-applied, pixel-level change-detection method [174]. For the different images, different methods are used to obtain the final change-detection results, including the thresholding method, pattern classification method, Markov random field method, multivariate statistical analysis method, and so on [175,176,177]. Due to the advantage of multi-level complex-feature extraction, end-to-end, pre-training, large-scale training sets, and other deep learning training mechanisms have also been applied to change detection [178].

The pixel-level change-detection method is easily affected by illumination difference, registration error, and noise. The image object contains more global information than a single pixel, and the object-based image analysis method is closer to the process of human-eye recognition. Object-level change detection is suitable for a comparative analysis of sustaining bodies of different meteorological disasters. According to the different change-detection strategies, the object-level change-detection methods can be roughly divided into direct object change detection, object change detection after synchronous segmentation, and object change detection after classification [179].

Scene change detection is the semantic level analysis of multi-time corresponding to the scene according to the semantic category of the changes and what changes have occurred, and applicable to the change detection of the overall environment of meteorological disasters. Compared with low-level analysis of the road changes, building changes and vegetation changes affected by meteorological disasters, it is better to analyze the disaster areas at the semantic level to provide information that conforms to the concept of human semantics. Remote sensing image scenes often contain a large number of different subject objects (buildings, vegetation, roads, etc.), but changes in the internal features of the scene cannot directly lead to changes in the type of scene. It can be concluded that there is a "spatio-temporal semantic gap" between the low-level changes in objects and the high-level changes in scene semantics. An instance of this is provided in Figure 4, which shows the evolution of the flooded area in the rice cultivations of the Albufera natural reserve (Spain) [180]. The purpose, in this case, is to highlight the presence of standing water and to provide an understanding of the scene land-cover through a balanced color-coding that is as close as possible to the natural color palette, which is the one the operator is used to.

Presently, pixel-level change detection is mostly used to detect various kinds of land surface indexes, object-level change detection is used to detect the damage degree of buildings, and scene-level change detection is used for flood mapping (Table 12).

#### 3.3.3. Change Detection Applications

Monitoring of disaster occurrence area. The determination of the disaster occurrence area is based on the construction of a comprehensive disaster index, which considers the intensity of the causing factor, the damage degree of the disaster-bearing body, and the impact of the disaster. On the basis of multi-temporal comprehensive disaster index inversion using remote sensing images, on the one hand, the disaster occurrence area where the disaster index changes are obtained by comparing the pre-disaster and post-disaster images; on the other hand, by comparing the images of different times, the change process of the disaster occurrence area is simulated [188].

Monitoring the degree of disaster. The disaster degree is mainly expressed by disaster grade, building damage degree, casualty, economic loss, etc. [189]. Remote sensing data can directly estimate the damage caused by disasters, and use the proportion of the population and socio-economic background data to indirectly estimate the casualties and economic losses, and then carry out an analysis of the degree of disaster. Through a comparison between multi-temporal images, the differences in the disasters’ impact can be analyzed, and the evolution of the degree of disaster (enhancement, persistence, attenuation, etc.) can also be observed.

Disaster rescue progress monitoring. Restoration and reconstruction are based on a comprehensive assessment of disasters, and the transitional resettlement of the victims is carried out. According to the post-disaster recovery and reconstruction plan, housing construction, infrastructure construction, public service facilities construction, disaster prevention and mitigation, ecological restoration, and land-use recovery are carried out [190]. A comparison between multi-temporal images can be used to monitor the progress of the restoration and reconstruction of houses and infrastructure. It can also monitor the effects of restoration and reconstruction, evaluate the humanistic effects and ecological effects of restoration and reconstruction, and provide scientific data to support subsequent prevention and control planning.

## 4. Open Problems and Challenges

### 4.1. Theoretical Basis: Process-Oriented Monitoring Perspective

Different types of meteorological disasters have different causes and effects, but they will evolve with the changes in disaster-causing factors and disaster carriers. Effective monitoring of the evolutionary process of meteorological disasters can not only promptly obtain the real-time status of the current disasters but also provide an objective basis for the prediction and early warning of the next disaster. Based on the above analysis, most research focused on monitoring a specific phase or a certain disaster factor [191,192]. However, meteorological disaster has a complex composition. Each part of the meteorological disaster affects the other parts and causes a state change, and the combination of different states constitutes the process of meteorological disaster. Therefore, it is necessary to clarify the integrity of the evolution process in meteorological disaster monitoring and to conduct in-depth theoretical research from the perspective of the meteorological disaster process. Starting with the whole process of meteorological disasters, not only can the occurrence state of meteorological disasters be monitored at a certain timepoint, but the causes of disasters can be inverted, and the consequences of disasters can be predicted. More importantly, this can provide an important reference for the study of the formation mechanism of meteorological disasters. In the future, with the urgent need for sustainable development and disaster prevention and mitigation, comprehensive monitoring of the evolution process of meteorological disasters should be strengthened due to the complexity, comprehensiveness, and integrity of meteorological disaster systems.

Regarding the methods for the remote sensing monitoring of meteorological disasters, most are multiple static spatiotemporal observations using multi-temporal remote sensing data. At present, machine-learning regression methods have advantages in terms of time dynamics, such as artificial neural networks with hidden layers, which can predict the temporal and spatial changes in geographic phenomena. As a typical geographical phenomenon, meteorological disasters can be used to compensate for the differences between disaster monitoring and disaster evolution at different timescales. Machine learning methods can be used to extract relevant information about the development process from the ever-increasing meteorological disaster monitoring data stream. This can not only simulate the evolution process of meteorological disasters, but also gain a further understanding of the scientific problems of the meteorological disaster system. Together, the theory and method research into meteorological disaster process-monitoring needs a further breakthrough.

### 4.2. Data Source: Reasonable Sensor Planning Scheme

Based on the characteristics of meteorological disasters, the requirements of the monitoring cycle, monitoring type, and monitoring range for different types and periods of meteorological disasters are determined. On the one hand, the applicability of existing sensors to disaster monitoring is clarified. For example, space-based remote sensing, mainly consisting of polar-orbiting satellites and geostationary satellites, has a wide range of coverage, a strong capacity for repeated observations, and is suitable for all-weather monitoring and early warning of large-scale dynamic changes in the pregnant environment and causing factors; space-based remote sensing, which is mainly based on aircraft, unmanned aerial vehicles and aerostat, has strong mobility and high resolution and is suitable for the dynamic monitoring of sudden meteorological disasters and the rapid assessment of the physical quantity of losses in the disaster-bearing body in key areas. On the other hand, to evaluate the combined effect of different sensors to acquire data, sensor planning can be obtained quickly and accurately in the actual monitoring of meteorological disasters, and the effectiveness and feasibility of the planning scheme are also guaranteed.

With the rapid development of big data technology, the availability of various types of data has been continuously broken, creating conditions for the comprehensive monitoring of meteorological disaster processes. Most of the recorded remote sensing data regard the natural environment of the earth’s surface, but the perception of changes in the social environment is scarce, and there is a lack of semantic attribute information. The fusion of multi-source data is not limited to the remote sensing data, but needs to be combined with more types of information to make up for the shortcomings of remote sensing monitoring. Social media, such as Twitter, Facebook, and Weibo, has become an effective means for government departments to keep abreast of disaster developments, with its wide participation and multi-source communication channels [193]. In the process of meteorological disasters, the real-time disaster information contained in social media is of great significance for a timely response and assessment of disasters [194,195]. Consequently, the fusion of remote sensing data and social media data provides new ideas and methods for process-monitoring of meteorological disasters.

### 4.3. Information Management: Spatio-Temporal Dynamic Information Modeling

The purpose of meteorological disaster process-monitoring is to perform a semantic analysis of various types of disaster information in multi-temporal remote sensing data. The existing meteorological disaster monitoring, due to the decentralization of its tasks, makes it possible to obtain one-sided disaster information. It is necessary to construct a standardized meteorological disaster process-monitoring information model and organize all kinds of information obtained from meteorological disaster monitoring in a standardized and integrated way.

The objects of meteorological disaster monitoring include the interaction law, spatial distribution and evolution process of various phenomena in the disaster system, with specific spatio-temporal characteristics. In the field of geographic information science, the geographic, spatio-temporal modeling methods mainly include the temporal snapshot model, event spatio-temporal model and process spatio-temporal model. (1) The temporal snapshot model represents geographic phenomena with continuous dynamic changes as a set of discrete objects with time stamps, displaying time as a property of space [196]. As only the object information of the state of time is expressed, the reason for the change is not recorded, and it is difficult to represent the geographical phenomenon of continuous, gradual change [197]. (2) The event spatio-temporal model is based on time and can reflect the temporal changes in geographical phenomena in fixed spatial locations [198]. However, the description of the evolving relation of the continuous dynamic phenomena is not suitable to represent the geographical phenomena of continuous changes in the spatio-temporal aspects [199]. (3) In contrast, the process spatio-temporal model directly takes the process characteristics of geographical phenomena as the modeling object, and integrates the space, time and attributes into a unified body using a correlation mechanism [200]. This logically solves the consistent representation of geographical phenomena in space, time and attribute continuous changes.

The process-oriented spatio-temporal modeling method can provide an important reference for the meteorological disaster process-monitoring information model. Moreover, the evolution mechanism of meteorological disasters is also reflected in the spatio-temporal process of disasters. Therefore, for information modeling in meteorological disaster process-monitoring, it is necessary to use time and space as the framework. This can formally describe the disaster content and mutual relationship among various disaster elements to provide an in-depth analysis, forecasting and early warning, disaster prevention, and relief.

### 4.4. Application Service: Diversified Visualization Solutions

By abstracting the disaster state, which is complex and difficult to directly perceive, the information visualization methods can effectively express the change in the morphological characteristics and attribute information of the disaster elements. These methods also can directly and show live the influence degree and development trend of disasters, and help people acquire knowledge and discover patterns. In the present and future, it is important to realize the visual display of disaster monitoring information to assist in disaster simulation, analysis, and emergency rescue decision-making.

Due to the diversity and persistence of meteorological disasters, the interrelation of all kinds of disaster information is complicated. The visualization of meteorological disaster monitoring information not only needs to deal with disaster data from different sources and complex disaster elements, but also needs to look at the key information of different users at various stages to develop visualization tasks. In the existing visualization research, no complete and universal method can accurately the characteristics of all meteorological disaster scenes and meet the needs of visualization information in different stages. In summary, visualizing the meteorological disaster monitoring information and displaying the evolutionary process of meteorological disasters is an urgent problem to be solved.

## 5. Conclusions

The occurrence and decline of meteorological disasters occur in a cyclic evolution process, and the remote sensing monitoring of meteorological disasters is essential for disaster prevention, disaster control, disaster resistance, disaster relief, and other disaster management. As meteorological disaster monitoring has the characteristics of variable timescales, diverse monitoring objects, and complex monitoring content, it is extremely challenging to obtain information on meteorological disasters efficiently, accurately and comprehensively. Based on the analysis of the basic concepts and methodology of remote sensing monitoring in meteorological disasters, this paper reviews and studies the research status of remote-sensing monitoring for meteorological disasters from the perspective of different meteorological disaster remote sensing tasks. The reasonable and comprehensive determination of monitoring tasks is a prerequisite for accurate and effective access to disaster information. In compliance with the requirements of the monitoring task, based on the extraction of surface parameters, land-cover information extraction, target recognition and other remote sensing processing approaches, information on the pregnant environment, causes and disaster-bearing body is extracted. Furthermore, by detecting and analyzing the changes in multi-temporal meteorological disaster information, the comprehensive monitoring of meteorological disaster evolution processes, such as disaster occurrence scope, degree of disaster effects, and disaster relief progress is realized.

Meteorological disaster monitoring is based on the time characteristics of the disaster process and analyzes the state changes in disaster content in each disaster element under a certain timescale. Although many remote sensing approaches were successfully used for meteorological disaster monitoring, there are still gaps in process monitoring. In the future, from the perspective of the process of meteorological disasters, the sensor-planning method and the information model of multi-temporal, remote sensing data should be studied. The improvement in various methods such as disaster information extraction abilities, the effective integration of multi-source data, and monitoring information visualization must be emphasized. More importantly, it is necessary to thoroughly study the objective internal laws of various disaster objects and disaster contents in meteorological disasters over different timescales, to reveal the relationship and influence mechanism of various pieces of disaster information in the evolution of meteorological disasters. This not only conforms to the logic and sequence of human understanding of meteorological disasters, but also provides support for the in-depth analysis of the evolution mechanism of meteorological disasters.

## Figures and Tables

**Figure 1 ijerph-19-03701-f001:**
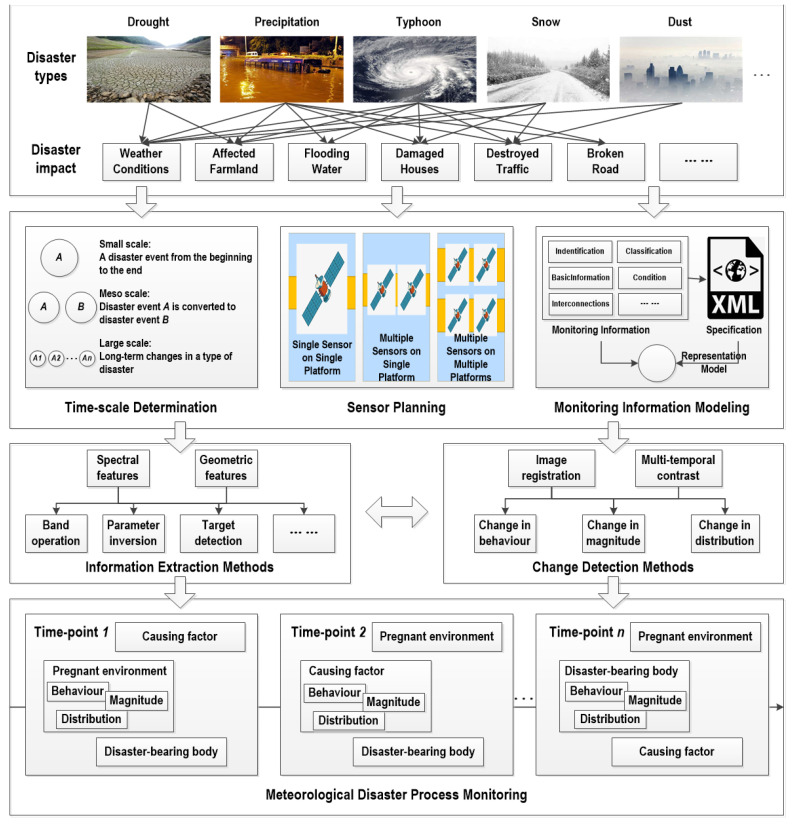
Technical roadmap of meteorological disaster monitoring based on remote sensing.

**Figure 2 ijerph-19-03701-f002:**
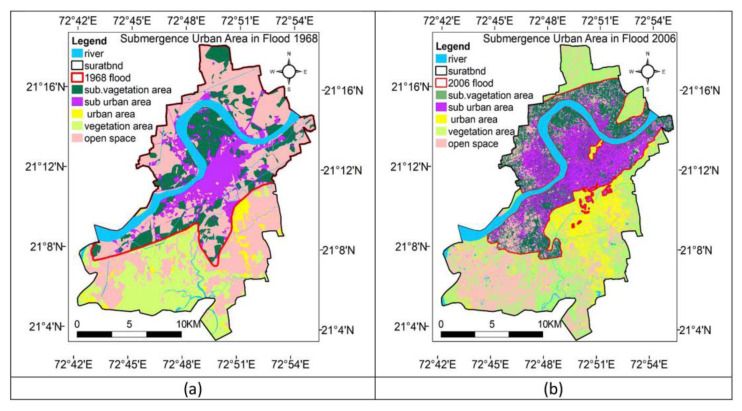
(**a**) Urban flood risk area of year 1968; (**b**) Urban flood risk area of year 2006 [81].

**Figure 3 ijerph-19-03701-f003:**
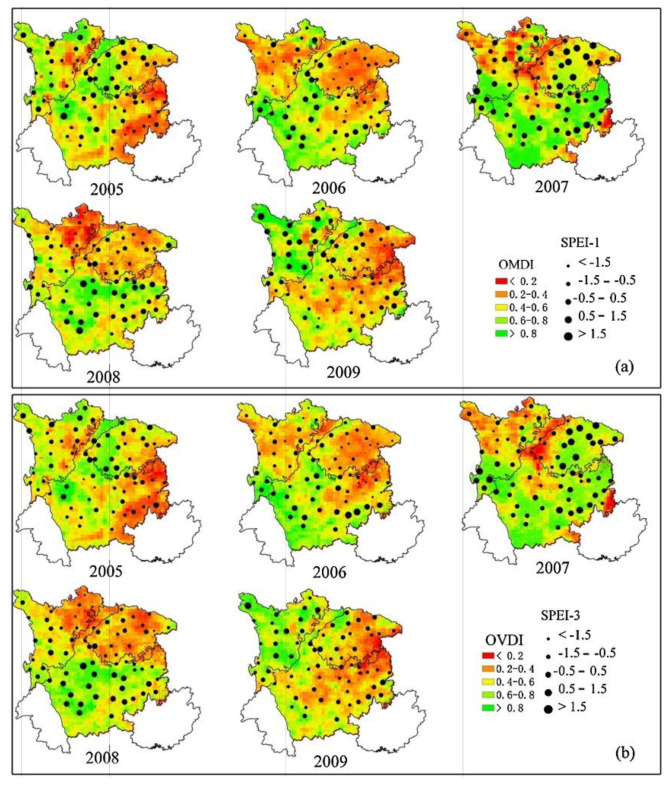
(**a**) Spatial distribution of drought monitored by OMDI with validations by SPEI-1and SPEI-3 in July between 2005 and 2009; (**b**) Spatial distribution of drought monitored by OVDI with validations by SPEI-1and SPEI-3 in July between 2005 and 2009 [91].

**Figure 4 ijerph-19-03701-f004:**
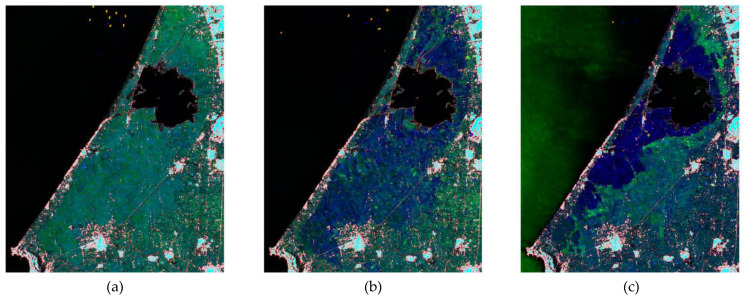
Evolution of the flooded area in the Albufera natural reserve (Spain). Pre-event image (blue band) acquired on 6 April 2017. Post-event images (green band) acquired on (**a**) 18 April 2017, (**b**) 4 August 2017 and (**c**) 14 December 2017 [180].

**Table 1 ijerph-19-03701-t001:** Appropriate observation timescale for different meteorological disaster elements.

Scale Classifications	Timescales			Applicable Types of Disaster Elements		
	Extent		Granularity (/y)	Pregnant Environment	Causing Factor	Disaster-Bearing Body
	Time Range(/y)	Timespan(/y)				
Extra-long scale	>100	>1000	100	Solar activity, climate change, landform change	-	Human, economic, environmental, social, and other major categories
		100–1000	10, 50, 100			
Long scale	10–100	50–100	10	Climate change, desertification, soil erosion	-	Human, economic, environmental, social, and other major categories
		10–50	1, 5, 10			
Meso scale	1–10	5–10	1	Land use, vegetation cover	Drought, precipitation, cold wave	Population (rural population, urban population, etc.), industry (heavy industry, light industry, etc.), agriculture (planting industry, breeding industry, etc.), construction (infrastructure, housing, etc.), transportation (land transport, water transport, etc.) and objects in the primary and secondary classifications in other categories.
		1–5	1 month, 1 year			
Short scale	≤1	1 year	10 days, 1 month, 1 season	-	Rainstorms, typhoons, high temperatures, cold waves, snowfall, hail, and other weather phenomena	Population (rural youth population, urban youth population, etc.), industry (food processing industry, metal manufacturing, etc.), agriculture (corn, wheat, etc.), construction (factory, community, etc.), transportation (railway, aircraft, etc.) and other objects in the primary, secondary, and tertiary classifications in other categories
		1 season	10 days, 1 month			
		10 days	1 day, 1 week			
		1 week	1 day			
		1 day	1 hour, 30 minutes, 1 minute			

**Table 2 ijerph-19-03701-t002:** Desired specifications of sensor focusing on meteorological disasters.

Types		Revisit Period (/d)	Spatial Resolution (/m)	Spectral Resolution (/nm)	Spectral Range (/μm)
Drought [45]		2–6	5–1000	2–50	0.76–14.0
Precipitation	Flood [46]	1–2	0.5–30	5–50	0.76–2.5
Typhoon [47]		2–12 h	50–1000	5–50	0.76–2.5
Snow	Snow cover [48]	1–2	5–30	2–50	0.76–2.5
	Sea ice [49]	2–5	30–1000	10–50	0.76–2.5
	Ice slush [50]	2–5	5–30	10–50	0.76–2.5

**Table 4 ijerph-19-03701-t004:** Information extraction for precipitation.

Type	Index/Method	Description
Cloud top radiation	Precipitation Estimation from Remotely Sensed Information using Artificial Neural Networks (PERSIANN) [92]	Cloud top radiation and reflection information are used to judge the possibility of precipitation. Determination of precipitation probability and duration from cloud thickness and cloud top temperature. The combination of the above precipitation estimates.
	Zhang et al., 2003 [93]	
Microwave-based inversion	Microwave radiation index [94]	Compared with infrared and visible light, microwaves can penetrate non-precipitation clouds; thus, it can detect the temperature and humidity information under all kinds of weather conditions except heavy precipitation.
	Microwave scattering index [95]	
	Comprehensive microwave index [96]	
Multi-sensor combination	TRMM microwave and TRMM rain radar [97]	Uses a combination of sensors to improve accuracy, coverage, and resolution.
	Microwave/infrared rainfall algorithm (MIRA) [98]	

**Table 5 ijerph-19-03701-t005:** Information extraction of snow.

Type	Index/Method	Description
Snow identification	Normalized difference snow index (NDSI) [99]	The inversion function is constructed by using the reflectivity of the green band and short-wave infrared band to distinguish snow cover from other ground objects and clouds.
	Normalized difference forest–snow index (NDFSI) [100]	
	Universal ratio snow index (URSI) [101]	
Snow coverage	MODSCAG algorithm [102]	Snow coverage can be estimated by the transition between snow-free and snow-free signals, which comes from the methods of mixed pixel decomposition and supervised classification.
	Sirguey et al., 2009 [103]	
	Hao et al., 2018 [104]	
	Aalstad et al., 2020 [105]	

**Table 6 ijerph-19-03701-t006:** Information extraction for haze.

Type	Index/Method	Description
Spectral feature	Lan et al., 1998 [106]	In the near-infrared band, haze particles are usually yellow or gray—white, while cloud and fog particles are usually white. In the thermal infrared band, the difference between haze and ground surface brightness temperature is small, but the difference between haze and cloud brightness temperature is large.
	Gao et al., 2015 [107]	
	Ge et al., 2017 [108]	
Image preferential	HOT transform [109]	The amplitude of the image after HOT transform is directly proportional to the influence degree of haze, and the value of HOT reflects haze’s degree of influence.
	Tasseled cap transformation [110]	The method is suitable for normalizing Landsat MSS with the influence of haze.
Aerosol optical depth	Li et al., 2013 [111]	The size of AOD was classified and the relationship between the degree of haze and the aerosol optical depth was analyzed. For instance, when AOD (440 nm) > 1, the haze occurred.
	Tao et al., 2014 [112]	
Multi-source remote sensing stereo monitoring	Optical sensor and laser radar [113]	Part of the solution to the lack of night observation data.
	Laser radar and infrared [114]	Distribution at different vertical heights.

**Table 7 ijerph-19-03701-t007:** Information extraction of typhoons.

Type	Index/Method	Description
Threshold range	Rau et al., 2013 [118]	The judgement is based on the different gray levels of different types of clouds in the same channel cloud image.
	Chen et al., 2015 [119]	
Mathematical morphology	Sasaki et al., 2015 [120]	The structure and characteristics of meteorological satellite cloud maps are described by using seven basic operations (dilation, erosion, opening, closing, hitting, thinning and thickening) in the space domain.
	Xie et al., 2017 [121]	
Dynamic clustering	Kitamoto et al., 2002 [122]	According to the principle of clustering, the sample points to be classified are clustered to the cluster center, and the iterative operation of modifying the cluster center is repeated.
	Liu et al., 2013 [123]	

**Table 8 ijerph-19-03701-t008:** Information extraction of dust.

Type	Index/Method	Description
Dark pixels	Kaufman et al., 1997 [125]	Based on the reflectance difference between the dust and the surface, the dust can be inverted according to the idea that the reflectance of the sand dust area is higher than that of the surface.
	Hsu et al., 2006 [126]	
Brightness temperature difference	Infrared difference dust index (IDDI) [127]	According to the difference between brightness temperature measured by remote sensing and surface brightness temperature in the clear sky, the occurrence of dust can be indicated to some extent.
	Infrared classification window algorithm [128]	
	Three channels [129]	
	Four channels [130]	

**Table 9 ijerph-19-03701-t009:** Different target types and applicable target detection methods.

Target Type	Typical Target	Target Signature	Main Detection Algorithm
Linear objects	Roads, airport runways, rivers, etc.	Edge, size, texture, grayscale	Data-driven strategies such as edge extraction, structural analysis, etc.
Blob objects	Trees, crops, buildings, vehicles, ships, etc.	Texture and shape features, spatial region features, visual saliency features	Data-driven strategies such as image segmentation, structural analysis detection, statistical analysis detection, distribution model (CFAR) detection, transform domain detection, multi-core learning, manifold learning, etc.
Complex objects	Residential areas, airports, stations, ports, etc.	Point, line, and texture features, multi-feature combinations	Task-driven and data-driven integrated strategy

**Table 10 ijerph-19-03701-t010:** Target detection approaches for disaster-bearing body.

Target Type	Index/Method		Description
Buildings	Image characteristics	Zhu et al., 2008 [135]	The characteristics in different images are compared and analyzed regarding gray-scale characteristics, shape characteristics, texture characteristics, and context characteristics.
		Nia et al., 2017 [136]	
		Ismail et al., 2022 [137]	
	Polarization characteristics	Ma et al., 2019 [138]	Polarization characteristics are sensitive to the shape and direction of target objects.
Roads	Image features	Boundary detection [139]	The image features are mainly the gray-level features, shape features, and edge features of the road.
		Morphological operator [140]	
		Binary conversion [141]	
	Dynamic programming	Poz et al., 2010 [142]	The parameter model of the road is transformed into the function of the road points, and the optimal path connecting the seed points of the road is determined.
		Fischler et al., 1987 [143]	
		Barzoha et al., 1996 [144]	
	Parallel line pair	Liang et al., 2008 [145]	The road edge has the characteristics of parallel line pairs.
Woodlands	Multiscale segmentation	Lai et al., 2022 [146]	Establish classification and recognition rule set to extract various damage information of natural forest.

**Table 11 ijerph-19-03701-t011:** Scene classification approaches for disaster-bearing body.

Scene Type	Index/Method		Description
Flood submerging area (Farmland)	Spectral signature	Li et al., 2014 [158]	Scene classification of water body remote sensing.
		Alfieri et al., 2014 [159]	
	Spatial shape and texture features	Henry et al., 2018 [160]	
		Li et al., 2015 [161]	
		Wang et al., 2017 [162]	
	Attention mechanism	Wang et al., 2016 [163]	Thematic information enhancement of submerged area based on selective visual attention mechanism.
		Wang et al., 2019 [164]	

**Table 12 ijerph-19-03701-t012:** The application of change detection.

Type	Index/Method	Description
Pixel-level change	Vegetation index [181]	Changes in agricultural drought distribution
	Haze index NDHI [182]	Changes in haze distribution
	Drought severity index (DSI) [183]	Changes in agricultural drought distribution
Object-level change	Strength characteristics [184]	Damaged condition of buildings
	Coherent characteristics [185]	
Scene-level change	Refice et al., 2014 [186]	Flood submerging area
	Cian et al., 2018 [187]	

## Data Availability

Not applicable.

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
