# Peer review of "Remote Sensing Approaches for Meteorological Disaster Monitoring: Recent Achievements and New Challenges"

_ijerph, 2022, doi:10.3390/ijerph19063701_

Round 1

Reviewer 1 Report

The paper reviews current research in the field of monitoring meteorological disasters. Based on the review, the author identifies open problems in the monitoring process and proposes the topics that future research may focus on.

I would suggest the author to consider the following points:

  • The text may result somewhat repetitive and overwhelming. I would suggest to review it and try to change some repetitive words (such as monitoring or meteorological disaster. For instance, lines 82-85: “For the remote sensing monitoring of meteorological disasters, the purpose is to meet the disaster management needs of disaster prevention and mitigation, and to obtain various disaster information in different periods in time. In essence, it is the application of remote sensing monitoring methods in the field of meteorological disasters”). I understand these words are key and are directly related to the topic; however, I think that their use in the text can be optimized in order to improve fluency.
  • Introduction: lacks a description of the structure of the paper at hands.
  • The author refers to remote sensing as if it was only image analysis (e.g. lines 125-126 or line 448). I suggest a wider allusion that includes other sensors and data, such as radar, LiDAR…
  • The roadmap of Figure 1 seems to be very relevant for this paper; however, little reference is done to it and a description of the roadmap is lacking.
  • Lines 181-182: author assures that “the difficulty in using multi-source sensor data is greatly increased, mainly reflected in the differences in data specifications (resolution, coordinate system, etc.), the inconsistency of the monitoring period, and the applicability of band fusion.” I respectfully disagree with that statement as uses the word “greatly”. Current data homogenization processes and data fusion strategies allows to successfully integrate different data sources. Please, consider changing or avoiding the word “greatly”.
  • Table 2. If the author created the content of table 2 based on his own knowledge, I suggest him to remark it. If he used previous work from other authors, then the references are missing.
  • Throughout paragraph “3. Remote sensing monitoring methods” I would like to find, not only the mention of previous studies, but also (and very importantly) their achievements, scope and accuracies. Or at least, those of the most relevant ones.
  • Lines 564-577. The author states that “Most existing methods for modeling disaster information (…) store time information and spatial information as common information elements” and claims that “it is necessary to use time and space as the framework, and can formally describe the disaster content…”. I would suggest to refer to spatio-temporal databases, since models for data storage and management that consider both, time and space components are available.
  • Some references are missing in the field of social media data integration, such as: https://www.mdpi.com/2071-1050/12/10/4246; https://ieeexplore.ieee.org/stamp/stamp.jsp?tp=&arnumber=6970293

Other minor issues:

  • I miss any reference to the UN COP.
  • Line 73: consider changing the rod “things” by a more meaningful one.
  • Line 459-460: Consider rephrasing.
  • Line 498-499: correct repetition.

Author Response

Response to Reviewer 1 Comments

Point 1. The text may result somewhat repetitive and overwhelming. I would suggest to review it and try to change some repetitive words (such as monitoring or meteorological disaster. For instance, lines 82-85: “For the remote sensing monitoring of meteorological disasters, the purpose is to meet the disaster management needs of disaster prevention and mitigation, and to obtain various disaster information in different periods in time. In essence, it is the application of remote sensing monitoring methods in the field of meteorological disasters”). I understand these words are key and are directly related to the topic; however, I think that their use in the text can be optimized in order to improve fluency.

Response 1: I thank the reviewer for the valuable comment. The duplicate words in the original sentence (lines 82-85) have been deleted. The modifying content of revised manuscript details refer to the lines 86-88 on the 2nd page. After carefully reading the full text, delete the duplicate concepts in the context of the manuscript.

Point 2. Introduction: lacks a description of the structure of the paper at hands.

Response 2: I agree with the reviewer's comment. This paper is focused on answering the following research questions:

(1) What are the objectives of meteorological disaster process monitoring?

(2) How are remote sensing approaches being used for meteorological disaster process monitoring?

(3) What are the current gaps in remote sensing-based meteorological disaster process monitoring?

Besides, I added the organization of this paper in revised manuscript, and the detailed revision can be found in lines 65-75 on the 2nd Page.

Point 3. The author refers to remote sensing as if it was only image analysis (e.g. lines 125-126 or line 448). I suggest a wider allusion that includes other sensors and data, such as radar, LiDAR…

Response 3: I agree with the reviewer's comment. The modifying content is that

…remote sensing data types, including optical image, radar, LiDAR and so on…

The modifying content of revised manuscript details refer to the lines 127-128 on the 3rd page. The "remote sensing image" in other sentences of the manuscript is also replaced or deleted according to the meaning of the sentences.

Point 4. The roadmap of Figure 1 seems to be very relevant for this paper; however, little reference is done to it and a description of the roadmap is lacking.

Response 4: I thank the reviewer for the valuable comment. A new paragraph has been added to illustrate the various modules and implications of the roadmap.

The modifying content of revised manuscript details refer to the lines 123-127 on the 3rd page.

Point 5. Lines 181-182: author assures that “the difficulty in using multi-source sensor data is greatly increased, mainly reflected in the differences in data specifications (resolution, coordinate system, etc.), the inconsistency of the monitoring period, and the applicability of band fusion.” I respectfully disagree with that statement as uses the word “greatly”. Current data homogenization processes and data fusion strategies allows to successfully integrate different data sources. Please, consider changing or avoiding the word “greatly”.

Response 5: I agree with the reviewer's comment. The modifying content is that

…the difficulty in using multi-source sensor data will increase to a certain extent…

The modifying content of revised manuscript details refer to the lines 190-192 on the 6th page.

Point 6. Table 2. If the author created the content of table 2 based on his own knowledge, I suggest him to remark it. If he used previous work from other authors, then the references are missing.

Response 6: Researchers are gradually focusing on sensor requirements for different meteorological disasters, analyzing the applicability of sensors from three aspects: temporal, spatial and spectral resolution. This paper is summarized in Table 2 according to the existing research results. The newly added references are marked.

The modifying content of revised manuscript details refer to the line 206 on the 6th page.

Point 7. Throughout paragraph “3. Remote sensing monitoring methods” I would like to find, not only the mention of previous studies, but also (and very importantly) their achievements, scope and accuracies. Or at least, those of the most relevant ones.

Response 7: I thank the reviewer for the valuable comment. The Section3 content has been heavily supplemented. On the one hand, the reference literature has been added and the content description is more detailed; On the other hand, it supplements the cases and explanations of the existing achievements, which makes the discussion more abundant.

The modifying content of revised manuscript details refer to the lines 245-576 on the 7th page-18th page.

Point 8. Lines 564-577. The author states that “Most existing methods for modeling disaster information (…) store time information and spatial information as common information elements” and claims that “it is necessary to use time and space as the framework, and can formally describe the disaster content…”.

Response 8: In the field of geographic information science, the modeling methods of geographic spatio-temporal mainly include temporal snapshot model, event spatio-temporal model and process spatio-temporal model. Therefore, the spatio-temporal modeling method can provide an important reference for the meteorological disaster process monitoring information model. Besides, the references of three kinds of modeling methods of of geographic spatio-temporal are supplemented in the manuscript.

The modifying content of revised manuscript details refer to the lines 649-666 on the 19th page.

Point 9. I would suggest to refer to spatio-temporal databases, since models for data storage and management that consider both, time and space components are available.

Some references are missing in the field of social media data integration, such as: https://www.mdpi.com/2071-1050/12/10/4246; https://ieeexplore.ieee.org/stamp/stamp.jsp?tp=&arnumber=6970293

Response 9: I thank the reviewer for the valuable comment. In the big data era, the real-time disaster information contained in social media is of great significance for timely response and assessment of disasters. Thus, the references of relevant researches have been added in this paper.

The modifying content of revised manuscript details refer to the lines 636-638 on the 19th page.

Point 10. I miss any reference to the UN COP.

Response 10: At the 26th Conference of the Parties to the United Nations Framework Convention on Climate Change, the World Meteorological Organization also issued the “State of the Global Climate 2021”, which identified extreme weather events as a major threat to global warming. The reference of UNCOP have been added in this paper.

The modifying content of revised manuscript details refer to the lines 58-62 on the 2nd page.

Point 11. Line 73: consider changing the rod “things” by a more meaningful one.

Response 11: I agree with the reviewer's comment. The modifying content is that

…Disaster is a general term for objects that…

The modifying content of revised manuscript details refer to the line 77 on the 2nd page.

Point 12. Line 459-460: Consider rephrasing.

Response 12: I agree with the reviewer's comment. The modifying content is that

…Due to the advantage of multi-level complex feature extraction, end-to-end, pre-training, large-scale training sets and other deep learning training mechanisms have also been applied to change detection…

The modifying content of revised manuscript details refer to the lines 516-519 on the 16th page.

Point 13. Line 498-499: correct repetition.

Response 13: The original description is cumbersome and repetitive, then in the revised version has removed this sentence.

The modifying content of revised manuscript details refer to the line 566 on the 18th page.

Reviewer 2 Report

This manuscript examines the utility of remote sensing applications for meteorological disasters. It provides a literature review of previous work accompanied by a speculative discussion of what may transpire in the future. I have specific comments that I identify below, but my main concern is this manuscript does not provide detail showing specific applicability of remote sensing to different types of meteorological disasters. It lacks practical applications and specific examples that are commonly found in the narratives of most journal articles with a purpose of providing a review of previous work. There are sections that provide a review of remote sensing components (image registration and change detection, for example), but then do not significantly examine how they are relevant to meteorological disaster monitoring or mitigation. A discussion of examples is needed throughout a reframed manuscript.

There are many book chapters and journal articles that touch on some of the content you cover here. You may want to review/include work by: Van Westen 2000, 2002; Rochon et al., 2008; Kaku, 2019; Boccorado and Tonolo, 2014; Im et al., 2019; Faisal et al., 2018.

Additionally, the manuscript has grammar errors and portions of it are awkwardly constructed. These issues include tense issues, missing words/articles, unclear phrasing, and repeated statements. The manuscript would benefit from copyediting and being reframed more concisely.

Lines 32-33: Are humans really able to prevent and/or control most natural disasters, especially meteorological disasters? Perhaps “mitigate” would be a more accurate descriptor.

Lines 42-43: What do you mean by “disaster-related departments”?

Lines 47-48: What do you mean by “the promotion of relevant international organization”?

Lines 61-63: A source is needed for this assertion. Otherwise, this is speculation.

Lines 101-102: Awkwardly constructed; what do you mean by “pregnant environment”? You define it later in the paragraph, but this paragraph needs to be reworked to increase clarity and conciseness.

Line 106: What do you mean by “material culture cycle”?

Lines 115-117: Does this contradict your statement in the previous sentence about returning to a normal state? Is a normal or equilibrium condition achieved if after one meteorological event ends, the next one begins?

Line 117: What do you mean by “all kinds of elements”? Be more specific.

Lines 130-131: “At present, research on meteorological disaster monitoring has involved one or more parts . . .”

Figure 1: The arrows at the top of the figure connecting disaster domains to disaster performances is confusing a difficult to follow. Also what do the boxes with … … mean?

Lines 174-175: Can you provide an example to show how this is applicable to meteorological monitoring? What information is not monitored by a single sensor?

Line 179: Again, make this relevant to meteorological monitoring. What types of information can be collected and why are some still limited?

Lines 180-181: Discuss the diverse information that can be monitored. Be specific.

Lines 251-253: Provide further explanation. Why?

Lines 275-276: This is another statement that needs a source. There are many natural environments that change rapidly, and others that may change over the time frames indicated, but have profound and even catastrophic impacts on humans. Also, what constitutes a “natural environment”? Is this sentence even necessary?

Line 280: “This research is not only conducive . . .”

Lines 279-283: This research is mentioned, but you do not explain or show how this work is a focus of pregnant environment monitoring.

Line 290: Why are spectral characteristics not discussed in the previous section regarding land cover?

Lines 296-301: How were these methods perfected? Explain how multi-source remote sensing data are used for drought monitoring.

Line 308: Why is radar remote sensing the main tool for short-term precipitation forecasting? You do not provide the same level of explanation with drought and rainstorm as you do the other disaster types.

Lines 330-331: Are there any examples of this?

Line 402: Source? Which studies have employed this?

3.3. Multi-temporal disaster information change detection: Much of this section presents a generalized review of image registration and change detection. If the purpose of the manuscript is to show the utility of remote sensing applications to meteorological disasters, the application of these processes should be the focus and deserve further development.

Lines 498-499: “the worst disaster area” is repeated

Line 517: “research” instead of “researches”

Lines 517-520: Sources are needed to support this statement. Provide examples.

Lines 568-571: Again, sources are needed. Provide examples.

4. Open problems and challenges: Portions of these sections are poorly constructed and written.

Lines 636-639: Sources are needed.

Author Response

Response to Reviewer 2 Comments

Point 1. This manuscript examines the utility of remote sensing applications for meteorological disasters. It provides a literature review of previous work accompanied by a speculative discussion of what may transpire in the future. I have specific comments that I identify below, but my main concern is this manuscript does not provide detail showing specific applicability of remote sensing to different types of meteorological disasters. It lacks practical applications and specific examples that are commonly found in the narratives of most journal articles with a purpose of providing a review of previous work. There are sections that provide a review of remote sensing components (image registration and change detection, for example), but then do not significantly examine how they are relevant to meteorological disaster monitoring or mitigation. A discussion of examples is needed throughout a reframed manuscript.

Response 1: I thank the reviewer for the valuable comment. In the revised version, extensive modifications have been made in two aspects. On the one hand, it refines the purpose of this review, and adjusts the content of the review according to this idea. Precisely, this paper is focused on answering the following research questions:

(1) What are the objectives of meteorological disaster process monitoring?

(2) How are remote sensing approaches being used for meteorological disaster process monitoring?

(3) What are the current gaps in remote sensing-based meteorological disaster process monitoring?

On the other hand, on the basis of supplementing relevant research literature, it adds the cases and explanations of the practical applications and specific examples, which makes the discussion more abundant.

Point 2. There are many book chapters and journal articles that touch on some of the content you cover here. You may want to review/include work by: Van Westen 2000, 2002; Rochon et al., 2008; Kaku, 2019; Boccorado and Tonolo, 2014; Im et al., 2019; Faisal et al., 2018.

Response 2: I thank the reviewer for the valuable comment. The recommended references were studied, and the references of relevant studies have been added in the revised version.

  1. Gamba, P.; Dell'Acqua, F.; Trianni G.; Stasolla M. GIS and Remote Sensing for Disaster Assessment in Urban Areas. In Pro-ceedings of 2007 Urban Remote Sensing Joint Event, Paris, France, 11-13 April 2007; pp. 1-5.
  2. Im, J.; Park, H.; Takeuchi, W. Advances in Remote Sensing-Based Disaster Monitoring and Assessment. Remote Sens. 2019, 11, 2181.
  3. Rochon, G.L.; Niyogi D.; Chaturvedi, A.; Arangarasan, R.; Madhavan, K.; Biehl, L.; Quansah, J.; Fall, S. Adopting Multisensor Remote Sensing Datasets and Coupled Models for Disaster Management. In: Nayak S., Zlatanova S. (eds) Remote Sensing and GIS Technologies for Monitoring and Prediction of Disasters. Environmental Science and Engineering (Environmental Science). Springer, Berlin, Heidelberg, 2008. doi: https://doi.org/10.1007/978-3-540-79259-8_5
  4. Faisal1, A.; Khan, H.A.H. Application of GIS and remote sensing in disaster management: a critical review of flood manage-ment. In Proceedings of International Conference on Disaster Risk Mitigation, Dhaka, Bangladesh, September 23-24, 2017.
  5. Van Westen, C. Remote sensing for natural disaster management. Int. Arch. Photogramm. Remote Sens. 2000, 33, 1609-1617.
  6. Kaku, K. Satellite remote sensing for disaster management support: A holistic and staged approach based on case studies in Sentinel Asia. Int. J. Disast. Risk Re. 2019, 33, 417-432.

Point 3. Additionally, the manuscript has grammar errors and portions of it are awkwardly constructed. These issues include tense issues, missing words/articles, unclear phrasing, and repeated statements. The manuscript would benefit from copyediting and being reframed more concisely.

Response 3: I agree with the reviewer's comment. I strengthened the revision manuscript for the article in the word and grammar proofreading, improve the writing quality of the paper.

Point 4. Lines 32-33: Are humans really able to prevent and/or control most natural disasters, especially meteorological disasters? Perhaps “mitigate” would be a more accurate descriptor.

Response 4: I agree with the reviewer's comment. The modifying content is that

…Mitigation of natural disasters are the key factor to ensure people's livelihood…

The modifying content of revised manuscript details refer to the lines 30-31 on the 1st page.

Point 5. Lines 42-43: What do you mean by “disaster-related departments”?

Response 5: The modifying content is that

…emergency management sectors…

The modifying content of revised manuscript details refer to the lines 40-41 on the 1st page.

Point 6. Lines 47-48: What do you mean by “the promotion of relevant international organization”?

Response 6: I agree with the reviewer's comment. Due to the ambiguity of the description, the phrase has been deleted in the revised manuscript.

Point 7. Lines 61-63: A source is needed for this assertion. Otherwise, this is speculation.

Response 7: At the 26th Conference of the Parties to the United Nations Framework Convention on Climate Change, the World Meteorological Organization also issued the “State of the Global Climate 2021”, which identified extreme weather events as a major threat to global warming. The reference of UNCOP have been added in this paper.

The modifying content of revised manuscript details refer to the lines 58-62 on the 2nd page.

Point 8. Lines 101-102: Awkwardly constructed; what do you mean by “pregnant environment”? You define it later in the paragraph, but this paragraph needs to be reworked to increase clarity and conciseness.

Response 8: I thank the reviewer for the valuable comment. The narrative mode of the paragraph is modified, and the repeated and wordy statements are deleted to increase the readability of the key contents.

For meteorological disasters, the pregnant environment refers to the comprehensive earth surface environment composed of the atmosphere, lithosphere and hydrosphere.

The modifying content of revised manuscript details refer to the lines 105-107 on the 3rd page.

Point 9. Line 106: What do you mean by “material culture cycle”?

Response 9: I agree with the reviewer's comment. Due to the ambiguity of the description, the phrase has been deleted in the revised manuscript.

Point 10. Lines 115-117: Does this contradict your statement in the previous sentence about returning to a normal state? Is a normal or equilibrium condition achieved if after one meteorological event ends, the next one begins?

Response 10: In the process of one meteorological disaster, the affected degree of the natural environment first aggravated, and then gradually restored to normal state. Moreover, when one meteorological disaster is over, the next one will start again, and the natural cycle repeats itself. The time scales of two meteorological disaster events and the process of one disaster are different, thus their states cannot be compared directly.

The modifying content of revised manuscript details refer to the lines 114-116 on the 3rd page.

Point 11. Line 117: What do you mean by “all kinds of elements”? Be more specific.

Response 11: All disaster factors (pregnant environment, causing factor and disaster-bearing body) are in the meteorological disaster process of mutual influence and constant change.

The modifying content of revised manuscript details refer to the lines 118-119 on the 3rd page.

Point 12. Lines 130-131: “At present, research on meteorological disaster monitoring has involved one or more parts . . .”

Response 12: I agree with the reviewer's comment. The chapter where the “Lines 130-131” are located has been revised to explain the composition and meaning of the roadmap. Therefore, the original sentence has been deleted in the revised version.

The modifying content of revised manuscript details refer to the lines 123-137 on the 3rd page.

Point 13. Figure 1: The arrows at the top of the figure connecting disaster domains to disaster performances is confusing a difficult to follow. Also what do the boxes with … … mean?

Response 13: The first line of the technical roadmap is the main type of meteorological disasters, and the second line is the potential destructive consequences of different meteorological disasters. The two lines are joined according to their correlation.

Remote sensing methods should not only pay attention to the causes of disasters, but also to the effects of disasters.

The “…” in the roadmap indicates that there are other similar types of information. However, due to the limit of the figure size, it is impossible to display all the contents in the roadmap.

Point 14. Lines 174-175: Can you provide an example to show how this is applicable to meteorological monitoring? What information is not monitored by a single sensor?

Response 14: On the basis of collecting related work, an example is added to illustrate the effect. For instance, Sivanpillai et al. studied rapid flood inundation mapping based on Landsat images [39]. In fact, there are many types of information that a single sensor can provide. Only when compared with multi-sensor, the information of single sensor is relative less.

  1. Sivanpillai, R., Jacobs, K.M., Mattilio, C.M., Piskorski, E.V. Rapid flood inundation mapping by differencing water indices from pre- and post-flood Landsat images. Front. Earth Sci. 2021, 15, 1–11.

The modifying content of revised manuscript details refer to the lines 175-176 on the 6th page.

Point 15. Line 179: Again, make this relevant to meteorological monitoring. What types of information can be collected and why are some still limited?

Response 15: On the basis of collecting related work, an example is added to illustrate the effect. For instance, the operational land imager (OLI) data and the thermal infrared sensor (TIRS) data were carried on Landsat-8. Kumar et al. derived the flooded areas from Landsat 8 OLI-TIRS image [40]. In fact, there are many types of information that multi-sensor in single platform can provide. Only when compared with multi-sensor in multiple platforms, the information of multi-sensor in single platform is relative less.

  1. Kumar, R.; Singh, R.; Gautam, H.; Pandey, M. Flood Hazard Assessment of August 20, 2016 Floods in Satna District, Madhya Pradesh, India. Remote Sens. Appl. Soc. Envron. 2018, 11, 104-118.

The modifying content of revised manuscript details refer to the lines 179-181 on the 6th page.

Point 16. Lines 180-181: Discuss the diverse information that can be monitored. Be specific.

Response 16: On the basis of collecting related work, an example is added to illustrate the effect. For instance, Kyriou et al. also studied flood mapping, while the approach is used us-ing data from active and passive remote sensing sensors like Sentinlel-1 and Landsat-8 [42]. In addition, the potential utility of multi-sensor data fusion for different phases of disaster management: vulnerability assessment, early warning systems, disaster miti-gation, response, damage assessment and recovery are delineated [43]. The relevant methods are in Section 3.2.2.

  1. Kyriou, A.; Nikolakopoulos, K. Flood mapping from Sentinel-1 and Landsat-8 data: a case study from river Evros, Greece. In Proceedings of Earth Resources and Environmental Remote Sensing/GIS Applications, Toulouse, France, 20 October 2015; doi: https://doi.org/10.1117/12.2194449
  2. Rochon, G.L.; Niyogi D.; Chaturvedi, A.; Arangarasan, R.; Madhavan, K.; Biehl, L.; Quansah, J.; Fall, S. Adopting Multisensor Remote Sensing Datasets and Coupled Models for Disaster Management. In: Nayak S., Zlatanova S. (eds) Remote Sensing and GIS Technologies for Monitoring and Prediction of Disasters. Environmental Science and Engineering (Environmental Science). Springer, Berlin, Heidelberg, 2008. doi: https://doi.org/10.1007/978-3-540-79259-8_5

The modifying content of revised manuscript details refer to the lines 185-189 on the 6th page.

Point 17. Lines 251-253: Provide further explanation. Why?

Response 17: This is because the empirical model is based on a large number of remote sensing data characteristics and their corresponding statistical results of ground realities. The difference in weather and insolation at different times affects the characteristics of remote sensing data. The difference of land type in different spaces affects the statistical results of ground realities. Therefore, the empirical model built under the specific space-time is not suitable for a large-scale extension.

The modifying content of revised manuscript details refer to the lines 265-270 on the 8th page.

Point 18. Lines 275-276: This is another statement that needs a source. There are many natural environments that change rapidly, and others that may change over the time frames indicated, but have profound and even catastrophic impacts on humans. Also, what constitutes a “natural environment”? Is this sentence even necessary?

Response 18: I agree with the reviewer's comment. Due to the ambiguity of the description, the phrase has been deleted in the revised manuscript.

Point 19. Line 280: “This research is not only conducive . . .”

Response 19: I agree with the reviewer's comment. Because the language in the revised version has been modified, the phrase has been deleted.

Point 20. Lines 279-283: This research is mentioned, but you do not explain or show how this work is a focus of pregnant environment monitoring.

Response 20: In recent years, with the acceleration of global urbanization, a great deal of infra-structure construction has rapidly changed the land cover in a few decades or even less time [79]. The impact of urbanization on the changing pregnant environment and even the occurrence of meteorological disasters has gradually become one of the focuses of pregnant environment monitoring. Urbanization and urban consumption produce large amounts of greenhouse gases, which upset the balance of the earth's ecosystem. This is a direct driver of climate change, contributing to global warming and to the deterioration of the pregnant environment in the ecosystem. Monitoring the pregnant environment under urbanization by remote sensing is an important reference to assess the risk of meteorological disasters [80]. For instance, Waghwala et al. assess flood risk based on the satellite image of Resources-1, then they proposed that a change from low urbanization to a high urbanization pattern is the main driver for increasing the flood risk [81].

  1. Shakya, A.K.; Ramola, A.; Kashyap, A.; Van Pham, D.; Vidyarthi, A. Susceptibility Assesment of Changes Developed in the Landcover Caused Due to the Landslide Disaster of Nepal from Multispectral LANDSAT Data. In: Singh P., Sood S., Kumar Y., Paprzycki M., Pljonkin A., Hong WC. (eds) Futuristic Trends in Networks and Computing Technologies. Springer, Sin-gapore, 2020.
  2. He, X.; Zou A. The Risk Assessment of Urban Flood Disaster Based on Elevation Ladder Data. Adv. Mater. Res. 2012, 446, 3046-3050.
  3. Waghwala, R.K.; Agnihotri, P.G. Flood risk assessment and resilience strategies for flood risk management: A case study of Surat City. Int. J. Disast. Risk Re. 2019, 40, 101155.

The modifying content of revised manuscript details refer to the lines 294-305 on the 8th page.

Point 21. Line 290: Why are spectral characteristics not discussed in the previous section regarding land cover?

Response 21: I agree with the reviewer's comment. On the basis of collecting related work, an example is added to illustrate the effect. The description of spectral characteristics of images used in monitoring has been supplemented and references have been marked.

The modifying content of revised manuscript details refer to the lines 285-287 on the 8th page.

Point 22. Lines 296-301: How were these methods perfected? Explain how multi-source remote sensing data are used for drought monitoring.

Response 22: This problem has been modified in two ways. On the one hand, increase the description of this problem. Considering the relationship between “vegetation index + land surface temperature” and drought. Because of the diversity of indexes, the inversion of these indexes usually depends on multi-source remote sensing data. On the other hand, mark the relevant references and list the existing research results in the revised version.

The modifying content of revised manuscript details refer to the line 326 on the 9th page.

Point 23. Line 308: Why is radar remote sensing the main tool for short-term precipitation forecasting? You do not provide the same level of explanation with drought and rainstorm as you do the other disaster types.

Response 23: Compared with infrared and visible light, microwave can penetrate non-precipitation cloud, thus it can detect the temperature and humidity information under all kinds of weather conditions except heavy precipitation.

In order to ensure the consistency of the narrative style of each part of this paper, the information extraction methods of different disaster types in Section 3.2.2 are re-integrated. At present, the remote sensing monitoring approaches for precipitation have been sorted out in Table 4.

The modifying content of revised manuscript details refer to the line 341 on the 11th page.

Point 24. Lines 330-331: Are there any examples of this?

Response 24: On the basis of collecting related work, an example is added to illustrate the effect. The modifying content of revised manuscript details refer to the line 374 on the 12th page.

Point 25. Line 402: Source? Which studies have employed this?

Response 25: On the basis of collecting related work, an example is added to illustrate the effect. For instance, Wang et al. choose the corresponding sensors to collect the information of disaster scene, and propose the multi-sensor information fusion strategies for flood and some other disaster [165].

  1. Wang, Y.; Qin, Y.; Li, Z.; Deng, L. Research on the Strategy of Multi Sensor Information Fusion in Coal Mine Disaster Scene. Appl. Mech. Materials 2012, 220, 1929-1933.

The modifying content of revised manuscript details refer to the lines 458-460 on the 15th page.

Point 26. Multi-temporal disaster information change detection: Much of this section presents a generalized review of image registration and change detection. If the purpose of the manuscript is to show the utility of remote sensing applications to meteorological disasters, the application of these processes should be the focus and deserve further development.

Response 26: Comprehensively sort out the existing change detection methods, and synchronously collect the meteorological disaster application scenarios related to the detection methods. On this basis, table 12, the meteorological disaster applications of change detection, is constructed. In addition, the illustration of typical cases is also added in the paper. An instance of this is provided in Figure 4, which shows the evolution of the flooded area in the rice cultivations of the Albufera natural reserve (Spain) [180].

  1. Amitrano, D.; Di Martino, G.; Iodice, A.; Riccio, D.; Ruello, G. A New Framework for SAR Multitemporal Data RGB Repre-sentation: Rationale and Products. IEEE Trans. Geosci. Remote Sens. 2015, 53, 117–133.

The modifying content of revised manuscript details refer to the lines 538-549 on the 17th page.

Point 27. Lines 498-499: “the worst disaster area” is repeated.

Response 27: I agree with the reviewer's comment. Because the language in the revised version has been modified, the phrase has been deleted.

Point 28. Line 517: “research” instead of “researches”.

Response 28: The modifying content is that

…most researches focused on…

The modifying content of revised manuscript details refer to the lines 583-584 on the 18st page.

Point 29. Lines 517-520: Sources are needed to support this statement. Provide examples.

Response 29: I agree with the reviewer's comment. The description of the original content is not accurate. A fresh explanation has been replaced in the revised version, and references have been added.

…Based on the above analysis, most researches focused on the monitoring of a specific phase or a certain disaster factor [191,192]…

The modifying content of revised manuscript details refer to the lines 583-585 on the 18st page.

Point 30. Lines 568-571: Again, sources are needed. Provide examples.

Response 30: I agree with the reviewer's comment. The description of the original content is not accurate. A fresh explanation has been replaced in the revised version.

…The existing meteorological disaster monitoring, because of the decentralization of its tasks, makes it possible to obtain disaster information is one-sided. It is necessary to construct a standardized meteorological disaster process monitoring information model and organize all kinds of information obtained from meteorological disaster monitoring in a standardized and integrated way…

The process-oriented spatio-temporal modeling method can provide an im-portant reference for the meteorological disaster process monitoring information model. The existing spatio-temporal modeling methods are summarized into three types, and relevant examples are marked by references.

The modifying content of revised manuscript details refer to the lines 641-674 on the 19st page.

Point 31. 4. Open problems and challenges: Portions of these sections are poorly constructed and written.

Response 31: I agree with the reviewer's comment. In the revised version, the Section 4 is re-integrated and written from the perspectives of theoretical basis, data source, information management and application service.

Point 32. Lines 636-639: Sources are needed.

Response 32: I agree with the reviewer's comment. The description of the original content is not accurate. A fresh explanation has been replaced in the revised version.

…Although many remote sensing approaches have been successfully used for meteorological disaster monitoring, there are still gaps in process monitoring…
